# Complete Remission of Associative Immune-Mediated Hemolytic Anemia in a Dog Following Surgical Resection of Intestinal Leiomyosarcoma

**DOI:** 10.3390/vetsci6020055

**Published:** 2019-06-13

**Authors:** Masashi Yuki, Eiji Naitoh

**Affiliations:** Yuki Animal Hospital, 2-99 Kiba-cho, Minato-ku, Aichi 4550021, Japan; nedved3311@gmail.com

**Keywords:** dog, immune-mediated hemolytic anemia, leiomyosarcoma

## Abstract

A twelve-year-old male castrated Chihuahua with a severe, microcytic, hypochromic, and nonregenerative direct antiglobulin test positive anemia characterized by marked spherocytosis was referred to the veterinary hospital. Abdominal ultrasound revealed a peritoneal mass of unclear origin. Transfusion, followed by mass resection, rapidly resolved the anemia without further immunosuppressive treatment. Histopathology confirmed extraluminal jejunal leiomyosarcoma. Multiple mechanisms, including immune-mediated destruction, likely contributed to the anemia. To the authors’ knowledge, this is the first report that describes the resolution of immune-mediated hemolysis in a dog after the removal of an intestinal neoplasm.

## 1. Introduction

Immune-mediated hemolytic anemia (IMHA) is one of the most common immune-mediated diseases in dogs [1]. The development of non-associative IMHA is associated with the production of antibodies specific to normal molecules on the surface of erythrocytes. Glycophorin, a glycoprotein that spans across the plasma membrane, has been proposed as one of the most common red blood cell (RBC) membrane antigens targeted by autoantibodies [2,3]. These antibodies activate the complement cascade, which results in intravascular RBC lysis or opsonize RBCs and facilitates phagocytosis by cells of the monocyte–phagocyte system in the liver and spleen, thus resulting in extravascular hemolysis [3].

Leiomyosarcoma is a slow-growing, locally invasive, and malignant tumor of smooth muscle origin which is typically slow to metastasize; it is reportedly the second-most common canine intestinal tumor and the most common intestinal sarcoma in dogs [4].

Associative IMHA has been associated with a number of underlying, complex processes. The pathogenesis of associative IMHA is complex. For instance, it is triggered by the modification of antigens and molecular mimicry of the RBC membrane, or it is associated with normal RBC membranes. These triggers activate T and B cells to produce antibodies [5,6,7]. Documented and hypothesized causes of associative IMHA include bacterial, viral, rickettsial, parasitic, protozoan, and neoplastic diseases [1]. However, it has not been established whether infectious diseases other than babesia and neoplastic diseases serve as triggers for IMHA [8]. There is a single case of IMHA associated with leiomyosarcoma in dogs, but the details are unclear [9].

In this study, we report on the resolution of immune-mediated hemolysis in a dog after the removal of an intestinal leiomyosarcoma.

## 2. Case Presentation

In Japan, there is no ethics committee available for private-practice animal hospitals. Nevertheless, this research was conducted according to the ethical codes of the Japan Veterinary Medical Association. The samples obtained in this study were used after obtaining written consent from the dog owner.

A twelve-year-old male castrated Chihuahua was presented to our veterinary hospital with a five-day history of lethargy, anorexia, and weakness. The dog was treated with ursodeoxycholic acid for a suspicion of hepatitis by the referring veterinarian (full records were not available for review). The dog had not previously received a heartworm prophylaxis or a combined vaccination, including for rabies, for more than a year.

On physical examination, his weight was 3.25 kg (7.15 lb), body temperature was 38.5 °C (101.3 °F), heart rate was 150 beats/min, and respiratory rate was 60 breaths/min. His general appearance was quiet, alert, and responsive. The mucous membranes were pale, and he experienced weakness while walking. Neurological examination, including postural reactions, spinal reflexes, and cranial nerve reflexes, revealed no abnormalities. Abdominal palpation revealed a mass measuring approximately 8 cm in diameter in the caudal abdomen.

Hemogram revealed severe anemia (packed cell volume (PCV), 7%; reference interval (RI), 37–55%), with decreased mean corpuscular volume (58.6 fL; RI, 60–77 fL) and corpuscular hemoglobin concentration (29.4 g/dL; RI, 32–36 g/dL). Anemia was severely nonregenerative (69,600 cells/μL; reticulocyte production index < 2), microchromic, and microcytic. Moderate leukocytosis (22,900 cells/μL; RI, 6000–17,000) and leukocytosis characterized by neutrophilia (21,068 cells/μL; RI, 3000–11,500; bands, 0 cells/μL; segs, 21,068 cells/μL) (Celltac alpha; Nihon Kohden, Tokyo, Japan) were also observed.

Blood smear analysis revealed mild anisocytosis, polychromatic erythrocytes, and spherocytosis (≥5 spherocytes/× 100 (oil immersion field)) (Figure 1A) [8,10]. No infectious organisms or neoplastic cells were found. These findings were confirmed by board-certified veterinary pathologists at commercial laboratories.

Autoaggregation, both macroscopic agglutination and as assessed by a saline agglutination test (4 drops of saline per 1 drop of blood), were negative. Biochemical analyses revealed abnormalities, including increased activities of alanine aminotransferase (ALT; 932 U/L; RI, 17–78), aspartate aminotransferase (>1000 U/L; RI, 17–44), alkaline phosphatase (408 U/L; RI, 47–254), and creatine phosphokinase (256 U/L; RI, 49–166), increased concentrations of urea (36.0 mg/dL; RI, 9.2–29.2) and C-reactive protein (CRP; 13.2 mg/dL; RI, 0–1.0), and decreased concentrations of total protein (5.2 g/dL; RI, 5.3–7.6), albumin (2.0 g/dL; RI, 2.6–4.0), glucose (60 mg/dL; RI, 75–128 mg/dL), creatinine (0.3 mg/dL; 0.4–1.4 mg/dL), and total calcium (8.9 mg/dL; 9.3–12.1 mg/dL) (FUJI DRI-CHEM 7000V; FUJIFILM Corporation, Tokyo, Japan). The in-house heartworm antigen test using immunochromatography was negative (CHW Ag test kit, KYOKUTO, Tokyo, Japan). All measurements were performed at our veterinary hospital. Urinalysis (USG and dipstick) revealed a specific gravity of 1.020 and weak proteinuria. No birilubinuria or hemoglobininemia were observed. The urine protein:creatinine ratio (UPC) (Monolis, Tokyo, Japan) was normal (0.13; RI, <0.5).

Thoracic radiography and echocardiography revealed no abnormalities, and abdominal radiographs revealed a mid-abdominal mass. Abdominal ultrasonography revealed a mass (approximately 8 cm in diameter) with mixed echogenicity. The origin of the mass could not be identified. Except for gall bladder sludge, there were no obvious abnormal findings in the liver. No intraperitoneal lymphadenomegaly and no peritoneal effusion were observed. Interpretation of radiographic and echocardiographic images was performed by a general practitioner. Additional tests, including the direct antiglobulin test (DAT) (Monolis, Tokyo, Japan) and polymerase chain reaction (PCR) were used to detect a panel of vector-borne hemopathogens (*Anaplasma* spp., *Babesia* spp., *Bartonella* spp., *Ehrlichia* spp., *Hepatozoon* spp., *Leishmania* spp., *Neorickettsia risticii*, and *Rickettsia rickettsii*) (IDEXX Laboratories, Tokyo, Japan), and antinuclear antibody (ANA; IDEXX Laboratories, Tokyo, Japan) analyses. The tests were performed in a commercial veterinary medical laboratory. A serological examination for detecting the vector-borne hemopathogens was not performed. DAT was positive at both 4 °C and 37 °C. Further, the PCR for both vector-borne pathogens and ANA was negative.

Based on these results, anemia, including associative IMHA complicated by chronic inflammation and bleeding caused by a tumor of unknown origin, was suspected. The intraperitoneal mass was suspected to have triggered IMHA, and thus surgical resection was planned. Whole blood transfusion (33.8 mL/kg) from a single appropriate donor, as identified based on blood type and results of cross-matching tests, was performed. Post-transfusion PCV increased to 29%. The glucose level and albumin concentration were 70 mg/dL and 2.1 g/dL, respectively. After the blood transfusion, the dog’s clinical condition was judged to be stable enough to undergo surgery.

On day 2, an additional blood test performed before surgery revealed normal levels of total bile acid (5.0 μmol/L; RI, 0–15.3) (IMMUNO AU10V; FUJIFILM Corporation, Tokyo, Japan), prothrombin time (8.1 s; RI, 7.4–8.4), activated partial thromboplastin time (17.1 s; RI, 12.0–24.0), fibrinogen concentration (185 mg/dL; RI, 150–350) (COAG2; Wako, Osaka, Japan), and increased D-dimer concentration (53.9 μg/mL; RI, <2.0) (SpeLIA; Precision System Science, Chiba, Japan). All examinations were performed in our veterinary hospital. Anesthesia for laparotomy was introduced using propofol (Propoflo; Zoetis, Tokyo, Japan) and was maintained using isoflurane (Isoflurane for animal, Intervet, Tokyo, Japan). Atropine (Atropine sulfate injection; VEDECO, Missouri, USA), midazolam (Dormicum; Astellas, Tokyo, Japan), and butorphanol (Vetorphale; Meiji, Tokyo, Japan) were administered as pre-medications. Physiological saline (Otsuka normal saline; Otsuka, Tokyo, Japan) supplemented with 5% glucose (Otsuka glucose injection; Otuka, Tokyo, Japan) was administered as a fluid during laparotomy. The intraperitoneal mass (9 × 7 × 5 cm) originated from the jejunum, protruding from the serosal surface (Figure 2a). Hemoperitoneum and bleeding from the mass were not observed (Figure 2a). There were no findings of metastasis to other organs and the macroscopic appearance of the liver was normal. The portion of the jejunum harboring the mass was resected, and end-to-end anastomosis was performed. Full-thickness biopsy of the proximal jejunum and liver were performed simultaneously. Joint aspiration was performed to rule out systemic lupus erythematosus (SLE) at the time of surgery and revealed normal fluid levels. Postoperatively, intravenous infusions of acetic acid Ringer’s fluids (Soldem 1; Termo, Tokyo, Japan), meloxicam (0.2 mg/kg (0.09 mg/lb), subcutaneous injection (SC) (Metacam; Boehringer Ingelheim, Tokyo, Japan) for pain relief, cefalexin (20 mg/kg (9.1 mg/lb), SC, q 12 h) (Cefalexin; Nichi-iko, Toyama, Japan) for prevention of postoperative infection, and dalteparin Na (Fragmin 100 U/kg (45.4 U/lb), SC, q 12 h) (Nichi-iko, Toyama, Japan) for prevention of thrombus were administered.

Up to day 7, PCV remained within the range of 25–28% (Figure 3). Clinical signs observed at the first clinical visit, such as lethargy, anorexia, and weakness, improved. The prothrombin time, activated partial thromboplastin time, and fibrinogen concentration were normal. The D-dimer concentration (1.73 μg/mL) (Figure 3) and glucose level (93 mg/dL) returned to normal. The administration of cefalexin (20 mg/kg, per os (PO), q 12 h) continued, while the administration of dalteparin Na (100 U/kg, SC, q 12 h) was switched to aspirin (0.49 mg/kg (0.22 mg/lb), PO, q 12 h) (Bufferin combination tablet A81; Eisai, Tokyo, Japan) and continued (Figure 3).

The histopathological examination of the surgically resected mass revealed a mild atypical monomorphic proliferation of non-epithelial cells, exhibiting a spindle shape with abundant eosinophilic cytoplasm and resembling smooth muscle. Tumor cells were classified as malignant based on anisonucleosis, the increased nuclear–cytoplasmic ratio, and the presence of prominent nucleoli (Figure 2c). These findings indicated leiomyosarcoma. The mass had developed in the extraluminal space of the intestinal tract (Figure 2b). However, there was no transmural invasion of the tumor in the mucosa. Moreover, no intestinal bleeding was detected, but the lymphatic vessels on the mucosal surface were dilated (Figure 2b). In the jejunum proximal to the tumor, lymphangiectasia without any inflammatory changes was observed. In the liver, multifocal necrotic foci with infiltration of neutrophils and macrophages and bleeding were observed. We could not confirm whether these histopathological findings were caused by transient hypoxia due to anemia or whether they existed previously. All pathological examinations and subsequent diagnoses were confirmed by board-certified veterinary pathologists at commercial laboratories.

On day 9, PCV was 29%, upon which the dog was discharged from the hospital. On day 16, PCV recovered to 40%. In addition, the results of DAT were negative at both 4 °C and 37 °C (Figure 3). Codocytes were observed but no spherocytes were observed (Figure 1B). The albumin level (2.9 g/dL) and CRP concentration (<0.9 mg/dL) recovered to normal. At this stage, all medications were discontinued. Activities of ALT (292 U/L) and aspartate aminotransferase (59 U/L) were mildly high; therefore, the available ursodeoxycholic acid was administered (10 mg/kg (4.5 mg/lb), PO, q 24 h) (Urso tablet; Tanabe Mitsubishi Pharmaceutical, Osaka, Japan). The patient was lost to follow-up, and thus it is unclear if liver values normalized.

## 3. Discussion

Reports indicate that in humans, associative IMHA associated with neoplastic disorders occurs in cases of renal cancer, ovarian cancer, thymic carcinoma, Kaposi sarcoma, breast cancer, pancreatic cancer, and prostatic cancer [11]. In dogs, associative IMHA associated with neoplastic disorders has been reported in cases of leukemia, lymphoma, gastric tumor, lung carcinoma, pancreatic carcinoma, duodenal leiomyosarcoma, splenic tumor, adrenal tumor, mastocytoma, undiagnosed abdominal tumor, reticulum cell sarcoma, chondrosarcoma, and poorly differentiated sarcoma [9,12,13,14,15]. However, few studies have been published that explain the progression of the disease and its treatment [14].

Although the mechanism of onset is complex for each case, the following mechanisms for IMHA associated with neoplastic disorders have been reported [6,7]: (1) the tumor modifies red cell antigens such that an autoantibody response is elicited, (2) autoantibodies against tumor antigens are produced, which cross-react with erythrocytes, (3) the tumor produces red cell autoantibodies, (4) immune complexes that are formed between tumor-associated antigens and antibodies bind to red cells and induce hemolysis through complement action, (5) immune dysfunction in patients with tumor results in the formation of forbidden clones of lymphocytes that frequently escape surveillance and are only recognized when they elicit clinical consequences, such as autoimmune hemolysis, and (6) failure of immune homeostasis occurs, which enables tumor growth and autoantibody production. Since several studies have reported that IMHA resolves with the disappearance of the underlying tumor lesion, surgical or medical treatment for the tumor is advised [5,6]. We hypothesized that a complete surgical resection of the antigenic tumor would inhibit the autoantibody production and that IMHA would spontaneously enter remission. Consistent with this hypothesis, a spontaneous remission occurred in the present case approximately 14 days after tumor resection. In dogs, immunoglobulin G is the primary autoantibody of IMHA [9,16,17], which has a half-life of approximately seven days [18]. The DAT was negative with no spherocytes seen on day 15 after the initial clinical visit. These findings are indicative of the absence of immunoglobulin G on RBC membranes after treatment.

In the present case, a diagnosis of IMHA was based on the presence of spherocytosis in combination with positive DAT. Spherocytes are present in the majority of dogs with IMHA and although the DAT has a low sensitivity for the diagnosis of IMHA, its specificity is high [8,10,19,20]. Spherocytes also appear in oxidative damage, envenomation, hypersplenism, hereditary erythrocyte membrane defects, and microangiopathic hemolytic disorders [8], but DAT is usually negative in these cases. Based on these factors, the present case was diagnosed as IMHA. Hemoglobinemia and jaundice were not observed, and erythroid regeneration was mild. These findings were different from typical findings of IMHA. Although the underlying causes for this discordance with typical IMHA findings could not be clarified, we hypothesized that the RBC count had already decreased due to anemia of chronic disease (ACD) with possible chronic intestinal bleeding. It is therefore possible that the number of RBCs destroyed by IMHA was small, and that this did not cause overt signs of hemolysis such as hyperbilirubinemia, bilirubinuria, or hemoglobinuria. Furthermore, because erythroid regeneration was mild, nonregenerative immune-mediated anemia and pure red cell aplasia were considered less likely, because spherocytes typically do not appear under such conditions. It was therefore considered more likely that the hemolysis had just begun. The authors believe it is most likely that multiple mechanisms, including immune-mediated destruction, contributed to the anemia. Because at the time of the initial clinical visit anemia was of the microcytic type, ACD due to existing leiomyosarcoma or hepatitis was suspected. Because hypoalbuminemia and an increased level of blood urea nitrogen were observed, chronic bleeding in the intestinal tract due to leiomyosarcoma was also suspected. However, no melena or findings suggestive of bleeding were observed on the histopathological examination. As no occult fecal blood test was performed, we cannot exclude the presence of very small traces of blood in the stool. Although iron deficiency anemia was suspected based on the presence of microcytosis and hypochromasia, a few thin rims of hemoglobin in RBCs were observed on the blood smear. Measurements of serum iron, total iron binding capacity, and unsaturated iron binding capacity were predicted to be useful, but severe anemia limited our ability to sample the blood.

In the present case, weak proteinuria and a high CRP concentration were observed at the initial clinical visit, and SLE was suspected. However, because ANA was negative and UPC and synovial fluid findings were normal, the possibility of SLE was ruled out. The hypoproteinemia and hypoalbuminemia initially were considered to be a consequence of gastrointestinal bleeding. However, signs such as melena or histopathologic evidence of transmural invasion of the tumor were not present. The presence of lymphangiectasia that was found on histopathology and considered to be a consequence of the physical obstruction of lymphatic vessels by the tumor could have caused some intestinal protein loss; however, other causes for hypoproteinemia such as chronic inflammation were also possible. In addition, hypoglycemia at the initial clinical visit recovered following the surgery. Sepsis and liver dysfunction were ruled out, and a typically reported tumor-associated syndrome due to leiomyosarcoma was suspected [21]. It was unclear whether abnormal pathological findings of the liver were caused by transient hypoxia due to anemia, or whether they existed previously. The patient was lost to follow-up; thus, the cause could not be identified. However, because IMHA was in remission after surgical resection of tumor, it is unlikely that the liver disease was associated with IMHA.

In the present case, microcytic hypochromic nonregenerative anemia with positive DAT and spherocytosis was resolved completely following the surgical resection of jejunal leiomyosarcoma, without the need for an immunosuppressive treatment. Furthermore, DAT was negative and spherocytosis completely resolved two weeks post-surgery. Therefore, anemia, including the associative IMHA complicated by chronic inflammation and gastrointestinal bleeding caused by the tumor, was suspected. To our knowledge, this is the first report that describes the resolution of immune-mediated hemolysis in a dog after the removal of an intestinal neoplasm without further need for an immunosuppressive treatment.

## Figures and Tables

**Figure 1 vetsci-06-00055-f001:**
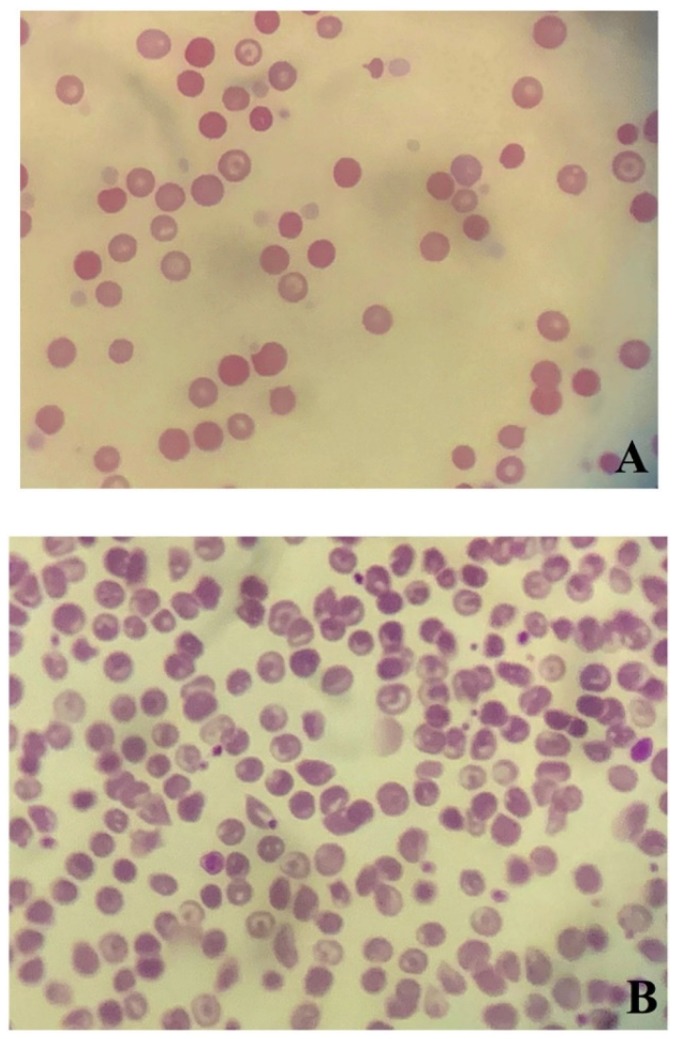
Blood smear findings (original magnification, 40×). (**A**) Mild anisocytosis, along with polychromatic erythrocytes and spherocytosis, were observed (initial clinical visit). (**B**) Codocytes were observed but no spherocytes were observed (day 16).

**Figure 2 vetsci-06-00055-f002:**
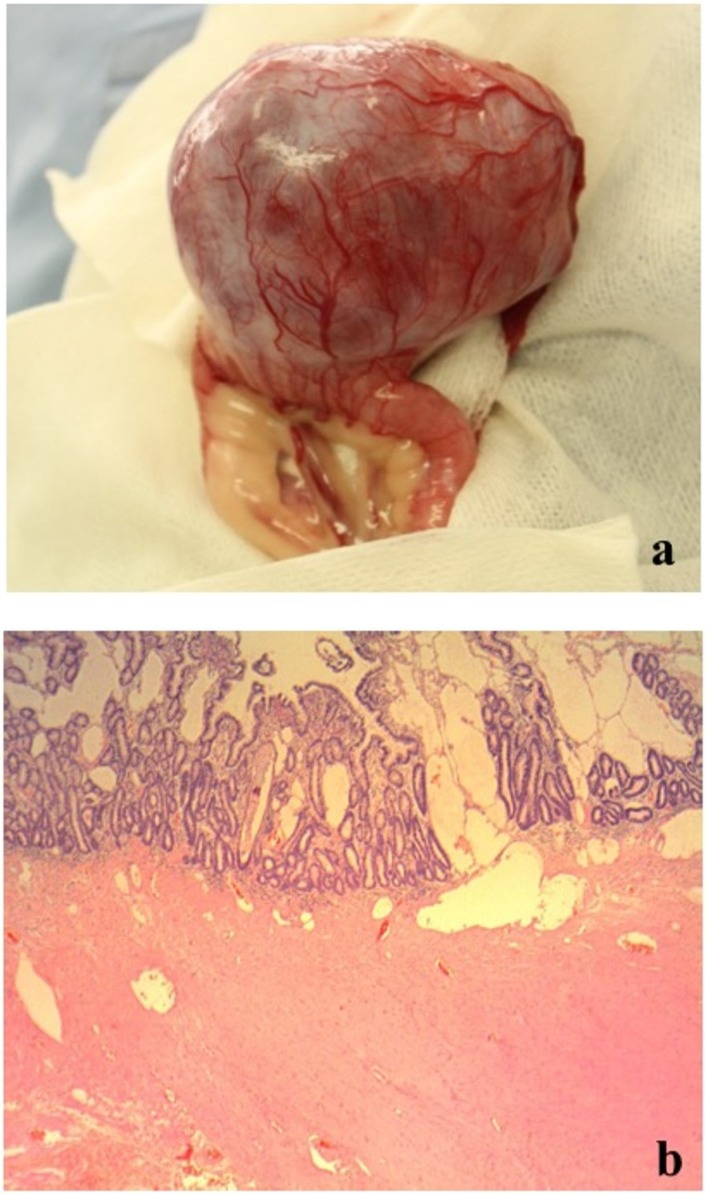
Macroscopic and histopathological findings of the surgically resected mass. (**a**) Macroscopic findings: The mass protruded from the serosal surface of the jejunum. (**b**) Histopathological findings (low-power field): The mass was located in the extraluminal space of the intestinal tract, but there was no transmural invasion of the tumor in the mucosa. No intestinal bleeding was detected, and lymphatic vessels on the mucosal surface were expanded. (**c**) Histopathological findings (high-power field): The mass revealed a mild atypical monomorphic proliferation of non-epithelial cells, exhibiting a spindle shape with abundant eosinophilic cytoplasm and resembling a smooth muscle. Tumor cells were classified as malignant based on anisonucleosis, the increased nuclear–cytoplasmic ratio, and the presence of prominent nucleoli.

**Figure 3 vetsci-06-00055-f003:**
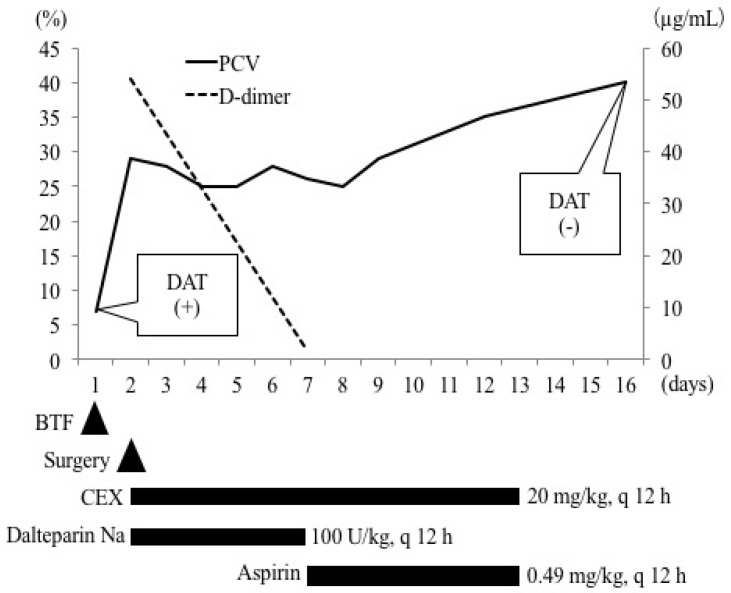
Results of changes in packed cell volume (PCV), Coombs test, D-dimer concentration, and administered drugs. DAT = direct antiglobulin test; BTF = blood transfusion; CEX = cephalexin.

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
