# Peer review of "Complete Remission of Associative Immune-Mediated Hemolytic Anemia in a Dog Following Surgical Resection of Intestinal Leiomyosarcoma"

_vetsci, 2019, doi:10.3390/vetsci6020055_

Round 1
Reviewer 1 Report
The authors reports that removal of a jejunal leiomyosarcoma leads to complete remission of Coomb’s positive anemia in a dog. To my knowledge, this has not been reported before. I think this is an interesting case, and should be published after revision (e.g., to simplify the Discussion section).
This is an interesting case report and should be published after the changes listed below:
Considering using Tables to present some of the results, e.g., those described on page 2.
Author Response
We are grateful to Reviewers 1, 2, and 3 for their critical comments and useful suggestions, which have helped to considerably improve our manuscript.
Responses to Comments from Reviewer 1
This is an interesting case report and should be published after the changes listed below:
Considering using Tables to present some of the results, e.g., those described on page 2.
Thank you for your critical review.
I have corrected manuscript with reference to all reviewers' comments.
We considered all the reviewer comments, and respectfully decided to not add a table.
Reviewer 2 Report
Brief summary
The article is overall well written and the overall message is clear. The presented case is an interesting case and therefore worthwhile to publish. However, I have some remarks in the work up of the case as some of the diagnostic tests that were done were not that important (or even unnecessary), while other more important tests were not performed. To increase the strength of this case presentation it is important that the discussion addresses these shortcomings. Overall, I think the case presentation is good, but the discussion is weak and needs to be improved.
Broad comments
The overall work up of the case was thorough and the interpretation of test results was fair. The case is complicated in a sense that it is difficult to say with certainty if this dog truly had immune-mediated hemolytic anemia (IMHA). In my opinion, this fact has not been discussed clear enough. Based on test results, it is my opinion, that it is highly likely that a component of the patient's anemia was caused by immune-mediated red cell destruction, but the challenges of diagnosing IMHA need to be discussed more thoroughly. I hereby refer to the ACVIM consensus guidelines regarding diagnosis of IMHA in dogs that were published in JVIM in March 2019. Based on these consensus guidelines, a high suspicion of IMHA is present when severe anemia is combined with at least 2 of the following (spherocytes >5/HPF; a positive direct anti globulin test and/or positive saline agglutination) in absence of another cause for the anemia. A definitive diagnosis of IMHA can only be made when the above is true in combination with at least 1 sign that is indicative of hemolysis (hyperbilirubinemia, bilirubinuria, hemoglobinemia or erythrocyte ghost cells). The authors mention briefly that no signs of hemolysis or hyberbilirubinemia were present, however, it is not mentioned if the urine was positive for bilirubin or hemoglobin and if there were ghost cells seen on the blood smear. This absolutely needs to be mentioned in the results. I assume that urine was negative for bilirubin and hemoglobin and that no ghost cells were seen on the blood smear. If this is the case, it needs to be more thoroughly discussed in the discussion that none of these were present. Furthermore, in dogs, spherocytes can provide evidence of hemolysis (I hereby refer references 65 and 66 from the ACVIM consensus guidelines). If no other indications for hemolysis were present, the authors need to discuss that spherocytes in dogs can be indicative of hemolysis. Differential diagnosis for spherocytes (such as hereditary erythrocyte membrane defects) need to be discussed and the authors need to mention that they considered other causes for the spherocytosis highly unlikely based on the age of the patient and the fact that the sperhocytosis resolved completely after resection of the intestinal mass.
Furthermore, the authors do not mention who interpreted the blood smear. Was this done by a board certified clinical pathologist? If the blood smear was interpreted by a general practitioner, it would be advised to send the blood smear (if still available) to a clinical pathologist to get a full report as ghost cells can easily be missed. Furthermore, it would increase the strength of the study if the blood smear was read by a specialist.
As for the other causes of the anemia: as the authors also report, it is clear to say that these dog had multiple causes for the anemia. However, not enough diagnostic tests were performed to actually exclude or include these other causes. The authors mention anemia of chronic disease as a more likely cause that gastro-intestinal (GI) blood loss. Unfortunately, no diagnostic tests were performed to evaluate for GI blood loss such as an occult faecal blood test. Furthermore, no tests were performed to evaluate the iron status in this patient. Ideally the iron status is based on evaluation of bone marrow stores of iron, but other tests such as ferritin or total binding iron capacity can give some indication on the iron status of a dog and can help us distinguish between functional iron deficiency (as expected with anemia of chronic disease) or absolute iron deficiency (as expected with chronic blood loss anemia). The authors could consider running these serological tests if serum of the patient was stored in the freezer. If this is not the case, the authors need to thoroughly discuss why these tests were not performed and need to discuss that it is considered highly likely that this patient had chronic GI blood loss, possibly in combination with anemia of chronic disease, but that this unfortunately was not fully documented.
Reasons for the anemia to be non regenerative in this case should also be more clearly discussed.First of all, with an absolute reticulocyte count of 69 000/microL, this dog is actually showing mild regeneration, as the authors also mention on line 191. Up to 30% of dogs with IMHA will present with non regenerative anemia, so it is not uncommon to see this. Secondly, based on the microcytic and hypochromic nature of the red blood cells, iron deficiency should be strongly suspected, which could further contribute to the anemia being non regenerative.
Further, it is not exactly clear why the authors are discussing the presence of proteinuria while they mention on line 81 that the urine protein:creatinin ration (UPC) was normal (the exact value of the UPC has not been reported). If the UPC was normal, all text regarding proteinuria and protein loss can be deleted as it is no contribution to this case. The presence of hypoalbuminemia and hypoproteinemia should be discussed, but is most likely due to chronic GI blood less, chronic inflammation (albumin is a negative acute phase protein), possibly some mild liver disfunction etc.
It is also not clear to me why there was a suspicion of SLE in this patient, as the only indication of immune-mediated disease was the suspicion of IMHA. On physical exam no abnormalities in gait or painful or effused joints were noted. I therefore am of the opinion that arthrocentesis was not indicated in this patient and would therefore likely not mention it in this report as it does not offer anything extra to this case.
In the case presentation, after line 82, all significant blood abnormalities (severe non regenerative microcytic hypo chromic anemia, moderate leucocytosis with neutrophilia, severe elevation of liver enzymes (ALT< AST, AP), elevated BUN urea with low creatinine, elevated CRP and moderate hypoalbuminemia and hypoproteinemia) should be discussed with for each abnormal finding the most important differential diagnosis. At the end of this discussion, the authors should then give a brief summary and explain what they considered was the most likely diagnosis in this patient and based on this what they decided to do next.
In the discussion it should be properly discussed how `associative IMHA` (or secondary IMHA) is usually treated. Often by removing the underlying cause, the IMHA will resolve spontaneously without need for further immunosuppressive treatment as was also the case for this patient. The authors should specifically discuss similarities and differences (in certain cases immunosuppressive treatment is also administered) between this case and previously reported case reports.
Titel
I would suggest a change to: Complete remission of suspected immune-mediated hemolytic anemia in a dog following surgical resection of intestinal leiomyosarcoma
Abstract
Line 10: replace Coomb`s by direct antiglobulin test (DAT)
Line 12: replace without immunosuppression by without further immunosuppressive treatment
Line 14: instead of `this is the first report ...`, write `To the author`s knowledge, this is the first report ...`
Introduction
Line 20: as suggested by the panel of the ACVIM consensus guidelines regarding the diagnosis of IMHA in dogs (JVIM, 2019) change `primary (idiopathic) IMHA` into `non-associative IMHA`
Line 25: add at the end of the sentence (resulting in extravascular hemolysis)
Line 26: change `secondary IMHA` into `Associative IMHA`
Line 26: replace `caused by` by `has been associated with a number of underlying processes`
Line 26: Before the sentence `For instance, ...` add `Its pathogenesis is complex`
Line 29-31: See ACVIM consensus guidelines and adjust paragraph: no strong causal link has been found for infections other than Babesiosis in dogs. Further, other than the causes the authors mention, generalized inflammatory conditions have been associated with IMHA, but again, no strong causal link has been established. The same goes for neoplastic conditions.
Case presentation
Line 36: Instead of `past medical history included hepatitis` it would be better to write `The dog was on treatment with ursodeoxycholic acid for a suspicion of acute hepatitis by the referring veterinarian (full records were not available for review).`
Line 37-38: remove sentence `hepatitis was diagnosed at another veterinary hospital and complete details of the diagnosis were not available for review`
Line 39: add if the dog had visited (or lived) in a region endemic for vector born diseases such as babesia, Ehrlichia, etc
Line 40: add what the general appearance of the dog was (BAR or QAR??)
Line 41: add if there were any abnormal auscultation findings such as a heart murmur, respiratory sounds etc
Line 44: add where the mass was felt: proximal, mid or caudal abdomen
Line 45: write immediately severe non regenerative microchromic microcytic anemia. And subsequently add all values (PCV, reticulocytes, MCV, MCH)
Line 45: change `pack` into `packed cell volume`
Line 46: add `moderate` before leucocytosis
Line 49-50: remove `the anemia was non regenerative and microcytic as you should mention this above`
Line 50-51: Report who evaluated the blood smear. Mention if erythrocytic ghosts were seen or any signs of oxidative red blood cell destruction (Heinz bodies, etc). Mention if infectious organisms or abnormal neoplastic cells were noticed.
Line 52: mention that both macroscopic autoagglutination as the saline agglutination test were negative.
Line 72: change `swelling of lymph nodes` into `intraperitoneal lymphadenomegaly`
Line 73-74: place the urinalysis findings immediately after the blood results (before medical imaging). And mention immediately the UPC value (no need to clarify that this test was done a day later, a brief mention of the normal UPC value and then the authors do not have to further address this). Mention if the urine had an abnormal macroscopic appearance (pigmenturia present?) and if it was positive for bilirubin
Line 75: change `coomb`s` into `DAT`. Explain briefly how this DAT was performed.
Line 77: Ehrichia is missing an L
Line 78: remove UPC as this should be reported with the urinalysis above
Line 81: remove `normal UPC was confirmed on the following day and negative for PCR and ANA` and just write `all other tests were negative`
Line 83: here should start a brief but thorough discussion of each of the important abnormalities on blood work (see above in general comments). The most important differential diagnosis should be discussed followed by a short summary of what the authors thought was most likely the cause for all the problems in this dog and a discussion of the further plan.
Line 84-85: remove `during the first clinical visit`
Line 86-87: remove `following the diagnosis of anemia including IMHA` as anemia is not a diagnosis but a clinical finding. The diagnosis at this point (presence of an intestinal leyomyosarcoma with secondary severe anemia) is not yet known.
Line 88: please be more specific how the patients condition was at this point `to a certain extent` is not specific enough. Report objective findings: overall presence, heart rate, respiratory rate etc
Line 89: this sentence `the intraperitoneal mass was suspected ....` should be removed from here and it should be placed before the blood transfusion after the discussion of what the authors thought was the cause of the findings in this dog.
Line 91-95: all these tests and results should be reported together with the additional tests on line 74. This allows the authors to also discuss abnormal findings with the discussion that should be provided (see Line 83). Were pre and postprandial bile acids measured? If not, discuss in discussion why this was not performed
Line 99: mitazolam needs a D instead of a T
Line 103: remove `of the jejunum` at the end of the sentence
Line 104: mention if there were any indications for metastasis. Also mention what the macroscopic appearance of the liver was.
Line 106-107: I would advise to remove that joint aspirations were performed as these were actually not indicated and do not provide any extra information in this case
Line 111: in the discussion the authors should very briefly discuss why fragmin was given as there is no published evidence that it prevents clot formation.
Line 113: change `symptoms` by `signs`
Line 117: the authors did not mention before that the dog was treated with aspirin. Please clarify.
Line 118: `was resumed`. Please clarify with which medications the dog was discharged from the hospital and for how long these meds were given. I see that this info was given in figure 3, but I would advise to write it in the text. Also clarify why the meds were given.
Line 25: rather than putting the `histopathological findings` between brackets, write them on line 122, just after B) Histopathological findings: The mass was located ...
Line 126: I would consider removing figure 3 as it is a bit confusing and I am unsure if it provides any added value
Line 130: remove `group` and change `the` by `a spindle shape`
Line 131: remove `and` and place a comma instead
Line 131-132: instead of `indicated` write `were indicative for` In the discussion the authors should discuss why the histopathologist concluded it was a leiomyosarcoma and not a leiomyoma as only mild atypical cells were seen. Any specific immunohistochemistry performed to distinguish? Perhaps state more clearly what indications for malignancy were detected on histopathology. It is very important to have a clear diagnosis of a sarcoma as the authors state this as the ultimate diagnosis in this patient
Line 134: add `but the lymphatic vessels on the mucosal surface were dilated`
Line 135: add `signs indicative for lymphangiectasia without inflammatory changes` and remove enteritis
Line 137: remove `however`
Line 142: replace `Coombs` by `DAT`
Line 143: other findings on blood smear? Anisocytosis, polychromasia etc??
Line 144: it would be interesting to know if these elevated liver enzymes eventually normalized? Any information about liver enzymes after day 9. If not, it should be stated, that the patient was lost to follow up and that it is therefore unclear if liver values normalized or not.
Line 145: very briefly discuss why just ursodeoxycholic acid was given and no other liver supportive treatment such as S adenosyl Methionine, milk thistle, vit E etc
Discussion
The authors are advised to take all the comments into account that are mentioned under `broad comments`
Line 155: change sentence to: `Although the mechanism of onset is considered complex for each case, following mechanisms for IMHA, associated with neoplastic disorders, have been reported:`
Line 164: here I would briefly discuss how `associative IMHA` (especially when associated with neoplastic disorders) is generally treated (based on previous case reports) and then the authors should discuss the similarities and differences between what other case reports report treatment wise and what they did and what the results were
Line 167: place the sentence `In dogs, immunoglobulin, ...` after the sentence that states that the Coombs was negative and no spherocytes were seen on day 15
Line 169: remove `in fact` and change `Coombs` into `DAT
Line 169: Not only DAT was negative, but blood smear was also negative for spherocytes on day 16. Mention both these findings. State clearly that this was an indication that immune mediated red cell destruction was considered to be absent at that time and that it is possible because immunoglobulin G .... with a half-life of .... was not present anymore
Line 169: remove the sentence `the number of days .... above hypothesis`
Line 171: now should follow a thorough discussion about diagnostic criteria of IMHA (based on ACVIM guidelines). Furthermore, it should be discussed why the authors had a suspicion of IMHA in this case. Also, it should be discussed that it is a limitation of the study that a full work up to determine the exact origin of the anemia was not performed (iron status in this dog? No occult blood dog) and the authors should discuss why these tests were not performed. Also, for other causes of the anemia (GI blood loss or anemia of chronic disease) it would also be expected that these would resolve with removing the intestinal tumor. Therefore, it is very important that the authors clearly discuss that they are aware of this and that the suspicion of secondary IMHA was based on the presence of spherocytes, the positive DAT and more importantly the disappearance of spherocytes on the blood smear on day 16 and the fact that the DAT became negative on day 14.
Line 171-182: I would consider deleting this section as it is not evidence based and can be confusing to readers
Line 183-185: see comment line 171. Mention that although the sensitivity of the DAT can be low, the specificity is high (94-100%) and that the authors therefore consider it unlikely that the DAT was a false positive results. For spherocytosis it should be discussed what the sensitivity and specificity of spherocytosis for IMHA is. Furthermore, the authors should discuss that spherocytes can also be present with other disorders (oxidative damage, envenomation, hypersplenism, hereditary erythrocyte membrane defects, microangiopathic hemolytic disorders) and should explain why they think the spherocytosis was likely due to immune mediated red cell destruction.
If blood smears were not evaluated by a clinical pathologist, this should be mentioned in the discussion as a limitation as it is possible that a general practitioner could have missed some other important findings on the blood smear.
Line 185-195: rewrite based on comments mentioned under `broad comments` and under Line 171
Line 195-198: discuss what could possibly the underlying reasons for the non regenerative nature of the anemia in this dog (30% of IMHA is non regenerative, iron deficiency, anemia of chronic disease and anemia due to chronic blood loss are typically non regenerative and the immune mediated destruction might only have been a small part of the anemia ...)
Line 199-209: see comment line 164 and discuss specifically how the treatment of this case compares with previous case reports
Line 205-209: are the authors aware of any specific studies that use (or evaluate) the use of human immunoglobulin in treatment of dogs with secondary or associative IMHA?? To my knowledge, studies have not shown that human immunoglobulin is an effective and advised treatment for IMHA in dogs. I would therefore suggest to remove this paragraph unless it is truly evidence-based
Line 210-212: remove sentences regarding SLE. There is more important stuff to discuss in the discussion
Line 212-216: a discussion of the possible causes for hypoalbuminemia should be done under case presentation (see comment line 83).
In the discussion at this point the authors should discuss the histopathology findings. Discuss the fact that the tumor did not invade the mucosa and that there were no signs of intraluminal bleeding. Briefly discuss the findings of the lymphangiectasia. The authors mention that this was likely a consequence of physical obstruction by the tumor. This is likely the case, but the authors should discuss why they believe this is the case (e.g. albumin levels (and also total protein??) normalized and the dog was not exhibiting any GI signs (I assume this is the case?) after surgery, no other signs indicative for chronic enteropathy on histopath etc)
Also the findings of histopathology of liver should be discussed here and the authors should conclude what they think is most likely the cause of these findings. For this it is important to know if the liver enzymes completely normalized eventually, if further liver dysfunction tests such as pre and post prandial bile acids were done etc ... If this was not the case, than the authors should state that the patient was lost to follow up
Line 219: replace Coombs by DAT
Line 219: write microcytic hypochromic non regenerative anemia with positive DAT and spherocytosis
Line 219: replace `resulted in complete remission of anemia` by `resolved completely
Line 220: behind `...surgical resection of jejunal leiomyosarcoma` add `without need of immunosuppressive treatment. Furthermore, DAT was negative and spherocytosis resolved completely two weeks post surgery.`
Line 221: replace `secondary` by `associative`
Line 221: add `gastro-intestinal` before bleeding
Line 222-223: I don't understand what the authors wish to say with `For IMHA cases, a careful diagnosis of its primary ..... this should be attempted first.` and would therefore advise to remove this sentence
Line 224: Add `To the author`s knowledge, this is the first report ....`
Line 225: Add `without further need of immunosuppressive treatment.`
Author Response
Responses to Comments from Reviewer 2
Brief summary
The article is overall well written and the overall message is clear. The presented case is an interesting case and therefore worthwhile to publish. However, I have some remarks in the work up of the case as some of the diagnostic tests that were done were not that important (or even unnecessary), while other more important tests were not performed. To increase the strength of this case presentation it is important that the discussion addresses these shortcomings. Overall, I think the case presentation is good, but the discussion is weak and needs to be improved.
I have corrected manuscript with reference to all reviewers' comments.
Broad comments
The overall work up of the case was thorough and the interpretation of test results was fair. The case is complicated in a sense that it is difficult to say with certainty if this dog truly had immune-mediated hemolytic anemia (IMHA). In my opinion, this fact has not been discussed clear enough. Based on test results, it is my opinion, that it is highly likely that a component of the patient's anemia was caused by immune-mediated red cell destruction, but the challenges of diagnosing IMHA need to be discussed more thoroughly. I hereby refer to the ACVIM consensus guidelines regarding diagnosis of IMHA in dogs that were published in JVIM in March 2019. Based on these consensus guidelines, a high suspicion of IMHA is present when severe anemia is combined with at least 2 of the following (spherocytes >5/HPF; a positive direct anti globulin test and/or positive saline agglutination) in absence of another cause for the anemia. A definitive diagnosis of IMHA can only be made when the above is true in combination with at least 1 sign that is indicative of hemolysis (hyperbilirubinemia, bilirubinuria, hemoglobinemia or erythrocyte ghost cells). The authors mention briefly that no signs of hemolysis or hyberbilirubinemia were present, however, it is not mentioned if the urine was positive for bilirubin or hemoglobin and if there were ghost cells seen on the blood smear. This absolutely needs to be mentioned in the results. I assume that urine was negative for bilirubin and hemoglobin and that no ghost cells were seen on the blood smear. If this is the case, it needs to be more thoroughly discussed in the discussion that none of these were present. Furthermore, in dogs, spherocytes can provide evidence of hemolysis (I hereby refer references 65 and 66 from the ACVIM consensus guidelines). If no other indications for hemolysis were present, the authors need to discuss that spherocytes in dogs can be indicative of hemolysis. Differential diagnosis for spherocytes (such as hereditary erythrocyte membrane defects) need to be discussed and the authors need to mention that they considered other causes for the spherocytosis highly unlikely based on the age of the patient and the fact that the sperhocytosis resolved completely after resection of the intestinal mass.
We have now revised the entire manuscript referring to your comments.
Furthermore, the authors do not mention who interpreted the blood smear. Was this done by a board certified clinical pathologist? If the blood smear was interpreted by a general practitioner, it would be advised to send the blood smear (if still available) to a clinical pathologist to get a full report as ghost cells can easily be missed. Furthermore, it would increase the strength of the study if the blood smear was read by a specialist.
We now list the clinical pathologist's smear findings and additional comments (lines 60–62).
As for the other causes of the anemia: as the authors also report, it is clear to say that these dog had multiple causes for the anemia. However, not enough diagnostic tests were performed to actually exclude or include these other causes. The authors mention anemia of chronic disease as a more likely cause that gastro-intestinal (GI) blood loss. Unfortunately, no diagnostic tests were performed to evaluate for GI blood loss such as an occult faecal blood test. Furthermore, no tests were performed to evaluate the iron status in this patient. Ideally the iron status is based on evaluation of bone marrow stores of iron, but other tests such as ferritin or total binding iron capacity can give some indication on the iron status of a dog and can help us distinguish between functional iron deficiency (as expected with anemia of chronic disease) or absolute iron deficiency (as expected with chronic blood loss anemia). The authors could consider running these serological tests if serum of the patient was stored in the freezer. If this is not the case, the authors need to thoroughly discuss why these tests were not performed and need to discuss that it is considered highly likely that this patient had chronic GI blood loss, possibly in combination with anemia of chronic disease, but that this unfortunately was not fully documented.
Reasons for the anemia to be non regenerative in this case should also be more clearly discussed.First of all, with an absolute reticulocyte count of 69 000/microL, this dog is actually showing mild regeneration, as the authors also mention on line 191. Up to 30% of dogs with IMHA will present with non regenerative anemia, so it is not uncommon to see this. Secondly, based on the microcytic and hypochromic nature of the red blood cells, iron deficiency should be strongly suspected, which could further contribute to the anemia being non regenerative.
We have now revised the entire manuscript referring to your comments.
We have now revised the text (lines 199-200).
Further, it is not exactly clear why the authors are discussing the presence of proteinuria while they mention on line 81 that the urine protein:creatinin ration (UPC) was normal (the exact value of the UPC has not been reported). If the UPC was normal, all text regarding proteinuria and protein loss can be deleted as it is no contribution to this case. The presence of hypoalbuminemia and hypoproteinemia should be discussed, but is most likely due to chronic GI blood less, chronic inflammation (albumin is a negative acute phase protein), possibly some mild liver disfunction etc.
UPC was measured only because it showed positive by dipstick. As you say, I think hypoproteinemia is another cause.
We have now revised the text (lines 212-215).
It is also not clear to me why there was a suspicion of SLE in this patient, as the only indication of immune-mediated disease was the suspicion of IMHA. On physical exam no abnormalities in gait or painful or effused joints were noted. I therefore am of the opinion that arthrocentesis was not indicated in this patient and would therefore likely not mention it in this report as it does not offer anything extra to this case.
The exclusion of immune-mediated polyarthritis is considered important in the differential diagnosis of elevated CRP, and polyarthritis may not show clinical symptoms. In addition, joint puncture under anesthesia does not burden the patient and thus we did not remove this finding.
In the case presentation, after line 82, all significant blood abnormalities (severe non regenerative microcytic hypo chromic anemia, moderate leucocytosis with neutrophilia, severe elevation of liver enzymes (ALT< AST, AP), elevated BUN urea with low creatinine, elevated CRP and moderate hypoalbuminemia and hypoproteinemia) should be discussed with for each abnormal finding the most important differential diagnosis. At the end of this discussion, the authors should then give a brief summary and explain what they considered was the most likely diagnosis in this patient and based on this what they decided to do next.
We have now revised the entire manuscript referring to your comments.
We have now revised the text (lines 163, 219-225).
In the discussion it should be properly discussed how `associative IMHA` (or secondary IMHA) is usually treated. Often by removing the underlying cause, the IMHA will resolve spontaneously without need for further immunosuppressive treatment as was also the case for this patient. The authors should specifically discuss similarities and differences (in certain cases immunosuppressive treatment is also administered) between this case and previously reported case reports.
We have now revised the entire manuscript referring to your comments.
We have now revised the text (lines 181-188).
Title
I would suggest a change to: Complete remission of suspected immune-mediated hemolytic anemia in a dog following surgical resection of intestinal leiomyosarcoma
Your suggestion is correct. However, we prefer to retain the term “associative IMHA” (line 2).
Abstract
Line 10: replace Coomb`s by direct antiglobulin test (DAT)
We have now revised the text (line 10).
Line 12: replace without immunosuppression by without further immunosuppressive treatment
This has been amended as requested (line 13).
Line 14: instead of `this is the first report ...`, write `To the author`s knowledge, this is the first report ...`
Thank you; this has been amended. (line 15).
Introduction
Line 20: as suggested by the panel of the ACVIM consensus guidelines regarding the diagnosis of IMHA in dogs (JVIM, 2019) change `primary (idiopathic) IMHA` into `non-associative IMHA`
We have now revised the text (line 21).
Line 25: add at the end of the sentence (resulting in extravascular hemolysis)
Thank you; we have now revised the text (line 27).
Line 26: change `secondary IMHA` into `Associative IMHA`
We have now revised the text (line 31).
Line 26: replace `caused by` by `has been associated with a number of underlying processes`
We have now revised the text (line 31).
Line 26: Before the sentence `For instance, ...` add `Its pathogenesis is complex`
We have now revised the text (lines 31-32).
Line 29-31: See ACVIM consensus guidelines and adjust paragraph: no strong causal link has been found for infections other than Babesiosis in dogs. Further, other than the causes the authors mention, generalized inflammatory conditions have been associated with IMHA, but again, no strong causal link has been established. The same goes for neoplastic conditions.
We have now revised the text in reference to your opinion (lines 35-37).
Case presentation
Line 36: Instead of `past medical history included hepatitis` it would be better to write `The dog was on treatment with ursodeoxycholic acid for a suspicion of acute hepatitis by the referring veterinarian (full records were not available for review).`
We have now revised the text (lines 43-44).
Line 37-38: remove sentence `hepatitis was diagnosed at another veterinary hospital and complete details of the diagnosis were not available for review`
We have now deleted this sentence.
Line 39: add if the dog had visited (or lived) in a region endemic for vector born diseases such as babesia, Ehrlichia, etc
We respectfully decline to add this information as we believe there is no need for this distinction.
Line 40: add what the general appearance of the dog was (BAR or QAR??)
We have added comments to expand on the appearance of the dogs (lines 48-49).
Line 41: add if there were any abnormal auscultation findings such as a heart murmur, respiratory sounds etc
There were no abnormal items to add.
Line 44: add where the mass was felt: proximal, mid or caudal abdomen
We have now revised the text (line 52).
Line 45: write immediately severe non regenerative microchromic microcytic anemia. And subsequently add all values (PCV, reticulocytes, MCV, MCH)
We have now revised the text (lines 54-56).
Line 45: change `pack` into `packed cell volume`
We have now revised the text (line 53).
Line 46: add `moderate` before leucocytosis
We have now revised the text (line 56).
Line 49-50: remove `the anemia was non regenerative and microcytic as you should mention this above`
We have deleted this sentence.
Line 50-51: Report who evaluated the blood smear. Mention if erythrocytic ghosts were seen or any signs of oxidative red blood cell destruction (Heinz bodies, etc). Mention if infectious organisms or abnormal neoplastic cells were noticed.
We now list the clinical pathologist's smear findings and additional comments (lines 60–62).
Line 52: mention that both macroscopic autoagglutination as the saline agglutination test were negative.
We have now revised the text (line 63).
Line 72: change `swelling of lymph nodes` into `intraperitoneal lymphadenomegaly`
We have now revised the text (line 85).
Line 73-74: place the urinalysis findings immediately after the blood results (before medical imaging). And mention immediately the UPC value (no need to clarify that this test was done a day later, a brief mention of the normal UPC value and then the authors do not have to further address this). Mention if the urine had an abnormal macroscopic appearance (pigmenturia present?) and if it was positive for bilirubin
We have now revised the text (lines 73-75). There was no abnormal macroscopic appearance found in urine.
Line 75: change `coomb`s` into `DAT`. Explain briefly how this DAT was performed.
We have now revised the text (line 87).
Line 77: Ehrichia is missing an L
We have now revised the text (line 89).
Line 78: remove UPC as this should be reported with the urinalysis above
We have now revised the text (lines 73–75).
Line 81: remove `normal UPC was confirmed on the following day and negative for PCR and ANA` and just write `all other tests were negative`
We have now revised the text (line 93).
Line 83: here should start a brief but thorough discussion of each of the important abnormalities on blood work (see above in general comments). The most important differential diagnosis should be discussed followed by a short summary of what the authors thought was most likely the cause for all the problems in this dog and a discussion of the further plan.
We have now revised the text (lines 94–97).
Line 84-85: remove `during the first clinical visit`
We have deleted this sentence.
Line 86-87: remove `following the diagnosis of anemia including IMHA` as anemia is not a diagnosis but a clinical finding. The diagnosis at this point (presence of an intestinal leyomyosarcoma with secondary severe anemia) is not yet known.
We have deleted this sentence.
Line 88: please be more specific how the patients condition was at this point `to a certain extent` is not specific enough. Report objective findings: overall presence, heart rate, respiratory rate etc
We have now revised the text (lines 99–100).
Line 89: this sentence `the intraperitoneal mass was suspected ....` should be removed from here and it should be placed before the blood transfusion after the discussion of what the authors thought was the cause of the findings in this dog.
We have now revised the text (lines 95–97).
Line 91-95: all these tests and results should be reported together with the additional tests on line 74. This allows the authors to also discuss abnormal findings with the discussion that should be provided (see Line 83). Were pre and postprandial bile acids measured? If not, discuss in discussion why this was not performed
I understand your opinion. However, because these test items were performed after whole blood transfusion, they did not change because they were not useful for diagnosis at the first examination. Bile acid was measured only preprandially.
Line 99: mitazolam needs a D instead of a T
We have now revised the text (line 109).
Line 103: remove `of the jejunum` at the end of the sentence
We have now revised the text (lines 112–113).
Line 104: mention if there were any indications for metastasis. Also mention what the macroscopic appearance of the liver was.
We have now revised the text (lines 114–115).
Line 106-107: I would advise to remove that joint aspirations were performed as these were actually not indicated and do not provide any extra information in this case
The exclusion of immune-mediated polyarthritis is considered important in the differential diagnosis of elevated CRP, and polyarthritis may not show clinical symptoms. In addition, joint puncture under anesthesia does not burden the patient and thus we did not remove this finding.
Line 111: in the discussion the authors should very briefly discuss why fragmin was given as there is no published evidence that it prevents clot formation.
We have fixed this mistake in the drug name now(line 122, Figure 3).
Line 113: change `symptoms` by `signs`
We have now revised the text (line 124).
Line 117: the authors did not mention before that the dog was treated with aspirin. Please clarify.
We have now revised the text (lines 128–129).
Line 118: `was resumed`. Please clarify with which medications the dog was discharged from the hospital and for how long these meds were given. I see that this info was given in figure 3, but I would advise to write it in the text. Also clarify why the meds were given.
We have now revised the text (lines 128–129).
Line 25: rather than putting the `histopathological findings` between brackets, write them on line 122, just after B) Histopathological findings: The mass was located ...
We have now revised the text (lines 133–140).
Line 126: I would consider removing figure 3 as it is a bit confusing and I am unsure if it provides any added value
Respectfully, we believe it is necessary to show the time-series of treatment and results, especially from surgery to the recovery of PCV.
Line 130: remove `group` and change `the` by `a spindle shape`
We have now revised the text (line 145).
Line 131: remove `and` and place a comma instead
We have now revised the text (line 146).
Line 131-132: instead of `indicated` write `were indicative for` In the discussion the authors should discuss why the histopathologist concluded it was a leiomyosarcoma and not a leiomyoma as only mild atypical cells were seen. Any specific immunohistochemistry performed to distinguish? Perhaps state more clearly what indications for malignancy were detected on histopathology. It is very important to have a clear diagnosis of a sarcoma as the authors state this as the ultimate diagnosis in this patient
The reviewer is correct, and thus we have added additional pathologist findings and added high-power field (lines 1347140, 146–147, Figure 2c).
Line 134: add `but the lymphatic vessels on the mucosal surface were dilated`
We have now revised the text (line 150).
Line 135: add `signs indicative for lymphangiectasia without inflammatory changes` and remove enteritis
We have now revised the text (line 151).
Line 137: remove `however`
We have removed this sentence (line 153).
Line 142: replace `Coombs` by `DAT`
We have now revised the text (line 158).
Line 143: other findings on blood smear? Anisocytosis, polychromasia etc??
We have now added our findings (lines 79-80, 159).
Line 144: it would be interesting to know if these elevated liver enzymes eventually normalized? Any information about liver enzymes after day 9. If not, it should be stated, that the patient was lost to follow up and that it is therefore unclear if liver values normalized or not.
We have now revised the text (lines 163–164).
Line 145: very briefly discuss why just ursodeoxycholic acid was given and no other liver supportive treatment such as S adenosyl Methionine, milk thistle, vit E etc
We have now revised the text (line 161).
Discussion
The authors are advised to take all the comments into account that are mentioned under `broad comments`
Line 155: change sentence to: `Although the mechanism of onset is considered complex for each case, following mechanisms for IMHA, associated with neoplastic disorders, have been reported:`
We have now revised the text (lines 173–174).
Line 164: here I would briefly discuss how `associative IMHA` (especially when associated with neoplastic disorders) is generally treated (based on previous case reports) and then the authors should discuss the similarities and differences between what other case reports report treatment wise and what they did and what the results were
We have expanded the literature in this regard to provide better context to our study (lines173–174).
Line 167: place the sentence `In dogs, immunoglobulin, ...` after the sentence that states that the Coombs was negative and no spherocytes were seen on day 15
We have now revised the text (line 188).
Line 169: remove `in fact` and change `Coombs` into `DAT
We have now revised the text (line 188).
Line 169: Not only DAT was negative, but blood smear was also negative for spherocytes on day 16. Mention both these findings. State clearly that this was an indication that immune mediated red cell destruction was considered to be absent at that time and that it is possible because immunoglobulin G .... with a half-life of .... was not present anymore
We have now revised the text (lines 188–189).
Line 169: remove the sentence `the number of days .... above hypothesis`
We have removed this sentence (lines 188–189).
Line 171: now should follow a thorough discussion about diagnostic criteria of IMHA (based on ACVIM guidelines). Furthermore, it should be discussed why the authors had a suspicion of IMHA in this case. Also, it should be discussed that it is a limitation of the study that a full work up to determine the exact origin of the anemia was not performed (iron status in this dog? No occult blood dog) and the authors should discuss why these tests were not performed. Also, for other causes of the anemia (GI blood loss or anemia of chronic disease) it would also be expected that these would resolve with removing the intestinal tumor. Therefore, it is very important that the authors clearly discuss that they are aware of this and that the suspicion of secondary IMHA was based on the presence of spherocytes, the positive DAT and more importantly the disappearance of spherocytes on the blood smear on day 16 and the fact that the DAT became negative on day 14.
We have rewritten this part of the discussion in consideration of this comment (lines 190–210).
Line 171-182: I would consider deleting this section as it is not evidence based and can be confusing to readers
We have removed this as suggested.
Line 183-185: see comment line 171. Mention that although the sensitivity of the DAT can be low, the specificity is high (94-100%) and that the authors therefore consider it unlikely that the DAT was a false positive results. For spherocytosis it should be discussed what the sensitivity and specificity of spherocytosis for IMHA is. Furthermore, the authors should discuss that spherocytes can also be present with other disorders (oxidative damage, envenomation, hypersplenism, hereditary erythrocyte membrane defects, microangiopathic hemolytic disorders) and should explain why they think the spherocytosis was likely due to immune mediated red cell destruction.
If blood smears were not evaluated by a clinical pathologist, this should be mentioned in the discussion as a limitation as it is possible that a general practitioner could have missed some other important findings on the blood smear.
Line 185-195: rewrite based on comments mentioned under `broad comments` and under Line 171
We have revised this part of the discussion in consideration of your opinion (lines 190–210).
Line 195-198: discuss what could possibly the underlying reasons for the non regenerative nature of the anemia in this dog (30% of IMHA is non regenerative, iron deficiency, anemia of chronic disease and anemia due to chronic blood loss are typically non regenerative and the immune mediated destruction might only have been a small part of the anemia ...)
We have revised this part of the discussion in consideration of your opinion (lines 190–210).
Line 199-209: see comment line 164 and discuss specifically how the treatment of this case compares with previous case reports
Line 205-209: are the authors aware of any specific studies that use (or evaluate) the use of human immunoglobulin in treatment of dogs with secondary or associative IMHA?? To my knowledge, studies have not shown that human immunoglobulin is an effective and advised treatment for IMHA in dogs. I would therefore suggest to remove this paragraph unless it is truly evidence-based
We agree with your opinion and have removed this section.
Line 210-212: remove sentences regarding SLE. There is more important stuff to discuss in the discussion
The exclusion of SLE and immune-mediated polyarthritis are important as differential diagnoses of elevated CRP; thus, we respectfully disagree with the reviewer and have retained this text in the manuscript.
Line 212-216: a discussion of the possible causes for hypoalbuminemia should be done under case presentation (see comment line 83).
In the discussion at this point the authors should discuss the histopathology findings. Discuss the fact that the tumor did not invade the mucosa and that there were no signs of intraluminal bleeding. Briefly discuss the findings of the lymphangiectasia. The authors mention that this was likely a consequence of physical obstruction by the tumor. This is likely the case, but the authors should discuss why they believe this is the case (e.g. albumin levels (and also total protein??) normalized and the dog was not exhibiting any GI signs (I assume this is the case?) after surgery, no other signs indicative for chronic enteropathy on histopath etc)
Also the findings of histopathology of liver should be discussed here and the authors should conclude what they think is most likely the cause of these findings. For this it is important to know if the liver enzymes completely normalized eventually, if further liver dysfunction tests such as pre and post prandial bile acids were done etc ... If this was not the case, than the authors should state that the patient was lost to follow up
Based on this constructive comment, we have modified the discussion accordingly. (lines 213–224).
Line 219: replace Coombs by DAT
We have now revised the text (line 227).
Line 219: write microcytic hypochromic non regenerative anemia with positive DAT and spherocytosis
We have now revised the text (line 227).
Line 219: replace `resulted in complete remission of anemia` by `resolved completely
We have now revised the text (line 228).
Line 220: behind `...surgical resection of jejunal leiomyosarcoma` add `without need of immunosuppressive treatment. Furthermore, DAT was negative and spherocytosis resolved completely two weeks post surgery.`
We have now revised the text (lines 229–230).
Line 221: replace `secondary` by `associative`
We have now revised the text (line 230).
Line 221: add `gastro-intestinal` before bleeding
We have now revised the text (line 231).
Line 222-223: I don't understand what the authors wish to say with `For IMHA cases, a careful diagnosis of its primary ..... this should be attempted first.` and would therefore advise to remove this sentence
We have removed this sentence as advised.
Line 224: Add `To the author`s knowledge, this is the first report ....`
We have now revised the text (line 232).
Line 225: Add `without further need of immunosuppressive treatment.`
We have now revised the text (lines 233–234).

Reviewer 3 Report
This manuscript describes a case of a dog with a primary leiomyosarcoma of the intestine showing a paraneoplastic immune-mediated hemolytic anemia (IMHA). This manuscript is interesting and can represent an original contribution for the Veterinary Literature. However, some information needs to be addressed before considering this manuscript for publication. The specific comments can be found below.
1. Abstract, page 1, lines 9-10. The first phrase is confusing and is probably lacking words. Suggestion for modification: “...A 12- year-old... by marked spherocytosis was referred to the Veterinary Hospital”.
2. Page 1, line 22, the word “RBC”. Please, spell acronyms before using the first time.
3. Introduction section, page 2. There is no information regarding intestinal leiomyosarcomas, or paraneoplastic immune-mediated hemolytic anemia (IMHA). Searching on PubMed the terms “paraneoplastic immune-mediated hemolytic anemia”, it is possible to find 41 items. Using “Intestinal leiomyosarcoma dog”, there are 17 items. Please, include a short description regarding intestinal leiomyosarcoma in dogs and IMHA as a paraneoplastic syndrome.
4. Case description page 1-2. The authors tried to establish a relation of IMHA and the intestinal leiomyosarcoma. However, the subject had a history of a previous hepatic disorder and there is no description of a clear timeline of this previous diagnosis. Can hepatitis or any hepatic disorder induce IMHA? Even without information regarding the previous diagnosis, I believe it is possible to establish a timeline between this previous hepatic disorder and the leiomyosarcoma diagnosis.
5. This report have a very important limitation: absence of information regarding this previous diagnosis. Some of the questions are: For this previous hepatic diagnosis, was ultrasound performed? Has this mass been identified at that time? Is there an association of this previous diagnosis with IMHA? The authors probably are not able to answer these questions, thus in my opinion, a statement of limitations and discussion of these questions should be included in the discussion section.
6. The figures are shown as A and B for figures 1 and 2. However, they seem independent figures (in different pages). Please, adjust.
7. Figure 2B. The histopathological image has a very low quality, out of focus and a lower power field that didn’t allow the reader to identify any neoplastic cell. Please, provide a new H&E staining image.
8. Fibrosarcoma is also a mesenchymal tumor that can develop in intestine. How to differentiate leiomyosarcoma and fibrosarcoma from histopathology? Was Masson's trichrome made? This reviewer believes that at least Masson's trichrome is essential for differentiation of both diagnosis.
9. Page 6, discussion section. Please, include in the discussion section a statement regarding the limitations of this case. Considering the comment “number 4”, if is possible to have a relation of the previous hepatic disease with the IMHA development, please, discuss as well.
Author Response
Responses to Comments from Reviewer 3
1. Abstract, page 1, lines 9-10. The first phrase is confusing and is probably lacking words. Suggestion for modification: “...A 12- year-old... by marked spherocytosis was referred to the Veterinary Hospital”.
We have now revised the text (line 11).
2. Page 1, line 22, the word “RBC”. Please, spell acronyms before using the first time.
We have now revised the text (lines 23-24).
3. Introduction section, page 2. There is no information regarding intestinal leiomyosarcomas, or paraneoplastic immune-mediated hemolytic anemia (IMHA). Searching on PubMed the terms “paraneoplastic immune-mediated hemolytic anemia”, it is possible to find 41 items. Using “Intestinal leiomyosarcoma dog”, there are 17 items. Please, include a short description regarding intestinal leiomyosarcoma in dogs and IMHA as a paraneoplastic syndrome.
We have now revised the text (lines 28-30, 37-38).
4. Case description page 1-2. The authors tried to establish a relation of IMHA and the intestinal leiomyosarcoma. However, the subject had a history of a previous hepatic disorder and there is no description of a clear timeline of this previous diagnosis. Can hepatitis or any hepatic disorder induce IMHA? Even without information regarding the previous diagnosis, I believe it is possible to establish a timeline between this previous hepatic disorder and the leiomyosarcoma diagnosis.
Thank you for the valuable comment. We have expanded the “case description” and “discussion” regarding this matter (lines 163, 219–226).
5. This report have a very important limitation: absence of information regarding this previous diagnosis. Some of the questions are: For this previous hepatic diagnosis, was ultrasound performed? Has this mass been identified at that time? Is there an association of this previous diagnosis with IMHA? The authors probably are not able to answer these questions, thus in my opinion, a statement of limitations and discussion of these questions should be included in the discussion section.
Thank you for this comment. We have expanded the “case description” and “discussion” to address this limitation (lines 163, 219–226).
6. The figures are shown as A and B for figures 1 and 2. However, they seem independent figures (in different pages). Please, adjust.
Thank you, we have now corrected this (Figure1, 2).
7. Figure 2B. The histopathological image has a very low quality, out of focus and a lower power field that didn’t allow the reader to identify any neoplastic cell. Please, provide a new H&E staining image.
Thank you, we have added additional pathologist findings and also provide a high-magnification picture (Figure 2c).
8. Fibrosarcoma is also a mesenchymal tumor that can develop in intestine. How to differentiate leiomyosarcoma and fibrosarcoma from histopathology? Was Masson's trichrome made? This reviewer believes that at least Masson's trichrome is essential for differentiation of both diagnosis.
Yes, we think you are correct on this. Accordingly, we have added additional pathologist findings and added high-power field (lines 146–147, Figure 2c).
9. Page 6, discussion section. Please, include in the discussion section a statement regarding the limitations of this case. Considering the comment “number 4”, if is possible to have a relation of the previous hepatic disease with the IMHA development, please, discuss as well.
We understand your opinion on this matter and have added additional information under “case description” and “discussion” (lines 163, 219–226).

Round 2
Reviewer 2 Report
It is clear to me that the author`s have put a lot of time and effort in to make all the necessary adjustments to the manuscript, for which I would like to thank them. I still have few minor remarks, but apart from these, I think this manuscript is worthwhile publishing.
Line 9: the Coombs test is actually the called the direct antiglobulin test so I would either write Coombs test (and than use Coombs test everywhere as you did previously) or write direct antiglobulin test (DAT) and than use DAT everywhere
Line 40: instead of `neoplasm` I would suggest to write `leiomyosarcoma`
Line 76: rather than saying that UPC was normal, I would give the value. Or say it was normal and write the value between brackets.
Line 86: lymphadenomegaly means swelling of the lymph nodes, so please remove `swelling of`
Line 90: Anaplasma is missing an `a`
Line 90: Ehrlichia is missing an `r`
Line 94: the way it is written now it seems like the DAT was negative for PCR and ANA. Make it two separate sentences: `DAT was positive at both ,,,, Further, PCR for vector-borne pathogens and ANA was negative`
Line 101: `to a certain extent` is too vague, rather write that after blood transfusion his clinical condition was judged to be stable enough to undergo surgery
Line 118: `systematic` should be `systemic`
Line 129: remove `from dalteparin Na to aspirin` in the middle of the sentence and write at the end of the sentence `Administration of cefalexin …. was continued and dalteparin Na was switched to aspirin …`
Line 141: it is `anisonucleosis` (please remove the `t`)
Line 141: `nucleolus` is missing an `l`
Line 148: instead of `because we observed` write `based on`
Line 148: it is `anisonucleosus` (please remove the `t)
Line 148: it should be `ratio` instead of `ration`
Line 148: in front of `nucleolus` (which is missing an `l`) write `the presence of prominent nucleoli` ==> it is the presence of prominent nucleoli that is indicative for malignancy
Line 164: as you are using `thus` also in the previous sentence, I would change `thus` for `therefore`
Line 183: I would change `since IMHA resolves` into `Since IMHA has been reported in some studies to resolve` ==> because there are not a lot of reports about IMHA and neoplasia so when you write `since IMHA resolves` it indicates that this is always the case, while there is not enough proof in literature to fully back up this statement
Line 184: I would change `performed` into `advised`.
Line 185: I would remove `based on these hypotheses` and I would change `proposed` into `hypothesized`
Line 186: change `will` into `would` both in middle of sentence and at the end of the sentence
Line 186-187: remove `as expected` as your hypothesis implies that you were expecting this
Line 190: change `are suggest` into `are suggestive for`
Line 191: write `In the present case, a diagnosis of IMHA was based on the presence of spherocytosis in combination with a positive DAT`
Line 191: Write `Spherocytes are present in the majority of dogs with IMHA and although the DAT has a low sensitivity for diagnosing IMHA, it`s specificity is high`
Line 194: `microangiopathy` should be `microangiopathic`
Line 194: instead of `DAT are not positive` write `but DAT is usually not positive in these cases`
Line 195: Please move the section 203-211 to here
Line 195: in front of `Anemia …` write `As anemia at the time of the ….
Line 196: remove `and`
Line 197: change `azotemia` into `an increased blood urea nitrogen`
Line 198: add `also` before suspected
Line 199: change `was` into `were`
Line 199: add `As no occult faecal blood test was performed, we cannot exclude the presence of very small traces of blood in the stool.`
Line 199: behind `although` add `Although, based on the presence of microcytosis and hypochromasia, iron deficiency ...
Line 203-211: I would move this whole section to just after `the present case was diagnosed as IMHA` on line 195
Line 201: remove `in addition`
Line 204: replace `however` by `Although reasons underyling this ,,, , we hypothesized that RBC count ,,,,`
Line 206: replace `in` by `due to`
Line 206: write `chronic intestinal bleeding` instead of `intestinal chronic bleeding`
Line 206: instead of `thus` write `it is therefore possible that the amount of RBCs destroyed by IMHA … and that this did not cause overt signs of hemolysis such as hyperbilirubinemia, bilirubinuria or hemoglobinuria`
Line 207: jaundice is the clinical syndrome that only is visible when bilirubin is markedly increased, so write hyperbilirubinemia instead of jaundice
Line 208: change `and` into `or`
Line 209: write `Pure red cell aplasia was however considered less likely because spherocytes ...
Line 210: remove `thus` and write `It was therefore considered more likely that the hemolysis had just begun`
Line 210: Write `The author`s think it is most likely that multiple mechanisms, including ...
Line 214-221: this section is just not strong enough and confusing. I think the author`s either need to rewrite it and thoroughly discuss all the possible differential diagnosis for hypoalbuminemia, hypoproteinemia and hypoglycemia or write something in the line of `The hypoproteinemia and hypoalbuminemia initially was thought to be a consequence of gastro intestinal bleeding. However, signs such as melena or histopathologic evidence of transmural invasion of the tumor were not present. The presence of lymphangiectasia that was found on histopathology and was thought to be a consequence of physical obstruction of the lymphatic vessels by the tumor, could have caused some intestinal protein loss, however other causes for hypoproteinemia such as chronic inflammation are also considered possible.` And then proceed with `In addition, hypoglycemia ….`
Line 222: write `due to anemia`
Line 223: remove `also`
Line 224-227: remove this section
Line 229: remove `was`
Author Response
We are grateful to Reviewer2 for their critical comments and useful suggestions, which have helped me to considerably improve our manuscript. As indicated in the responses that follow, we have taken all the comments and suggestions into account in the revised version of our manuscript.
Line 9: the Coombs test is actually the called the direct antiglobulin test so I would either write Coombs test (and than use Coombs test everywhere as you did previously) or write direct antiglobulin test (DAT) and than use DAT everywhere
We have revised the text (line 10).
Line 40: instead of `neoplasm` I would suggest to write `leiomyosarcoma`
We have revised the text (line 40).
Line 76: rather than saying that UPC was normal, I would give the value. Or say it was normal and write the value between brackets.
We have revised the text (line 75).
Line 86: lymphadenomegaly means swelling of the lymph nodes, so please remove `swelling of`
We have deleted this sentence (line 85).
Line 90: Anaplasma is missing an `a`
We have revised the text (line 88).
Line 90: Ehrlichia is missing an `r`
We have revised the text (line 89).
Line 94: the way it is written now it seems like the DAT was negative for PCR and ANA. Make it two separate sentences: `DAT was positive at both ,,,, Further, PCR for vector-borne pathogens and ANA was negative`
We have revised the text (lines 92-93).
Line 101: `to a certain extent` is too vague, rather write that after blood transfusion his clinical condition was judged to be stable enough to undergo surgery
We have revised the text (lines 99-100).
Line 118: `systematic` should be `systemic`
We have revised the text (line 117).
Line 129: remove `from dalteparin Na to aspirin` in the middle of the sentence and write at the end of the sentence `Administration of cefalexin .... was continued and dalteparin Na was switched to aspirin ...`
We have revised the text (lines 128-130).
Line 141: it is `anisonucleosis` (please remove the `t`)
We have revised the text (line 141).
Line 141: `nucleolus` is missing an `l`
We have revised the text (line 141).
Line 148: instead of `because we observed` write `based on`
We have revised the text (lines 147-148).
Line 148: it is `anisonucleosus` (please remove the `t)
We have revised the text (line 148).
Line 148: it should be `ratio` instead of `ration`
We have revised the text (line 148).
Line 148: in front of `nucleolus` (which is missing an `l`) write `the presence of prominent nucleoli` ==> it is the presence of prominent nucleoli that is indicative for malignancy
We have revised the text (line 148).
Line 164: as you are using `thus` also in the previous sentence, I would change `thus` for `therefore`
We have revised the text (line 163).
Line 183: I would change `since IMHA resolves` into `Since IMHA has been reported in some studies to resolve` ==> because there are not a lot of reports about IMHA and neoplasia so when you write `since IMHA resolves` it indicates that this is always the case, while there is not enough proof in literature to fully back up this statement
We have revised the text (lines 183-184).
Line 184: I would change `performed` into `advised`.
We have revised the text (line 184).
Line 185: I would remove `based on these hypotheses` and I would change `proposed` into `hypothesized`
We have revised the text (line 184).
Line 186: change `will` into `would` both in middle of sentence and at the end of the sentence
We have revised the text (lines 184-186).
Line 186-187: remove `as expected` as your hypothesis implies that you were expecting this
We have deleted this sentence (line 186).
Line 190: change `are suggest` into `are suggestive for`
We have revised the text (line 190).
Line 191: write `In the present case, a diagnosis of IMHA was based on the presence of spherocytosis in combination with a positive DAT`
We have revised the text (lines 192-193).
Line 191: Write `Spherocytes are present in the majority of dogs with IMHA and although the DAT has a low sensitivity for diagnosing IMHA, it`s specificity is high`
We have revised the text (lines 193-194).
Line 194: `microangiopathy` should be `microangiopathic`
We have revised the text (line 196).
Line 194: instead of `DAT are not positive` write `but DAT is usually not positive in these cases`
We have revised the text (lines 196-197).
Line 195: Please move the section 203-211 to here
We have revised the text (lines 198-208).
Line 195: in front of `Anemia ...` write `As anemia at the time of the ....
We have revised the text (line 208).
Line 196: remove `and`
We have deleted.
Line 197: change `azotemia` into `an increased blood urea nitrogen`
We have revised the text (line 210).
Line 198: add `also` before suspected
We have revised the text (line 211).
Line 199: change `was` into `were`
We have revised the text (line 212).
Line 199: add `As no occult faecal blood test was performed, we cannot exclude the presence of very small traces of blood in the stool.`
We have revised the text (lines 212-213).
Line 199: behind `although` add `Although, based on the presence of microcytosis and hypochromasia, iron deficiency ...
We have revised the text (lines 213-214).
Line 203-211: I would move this whole section to just after `the present case was diagnosed as IMHA` on line 195
We have revised the text (lines 198-208).
Line 201: remove `in addition`
We have removed.
Line 204: replace `however` by `Although reasons underyling this ,,, , we hypothesized that RBC count ,,,,`
We have revised the text (line 200).
Line 206: replace `in` by `due to`
We have revised the text (line 201).
Line 206: write `chronic intestinal bleeding` instead of `intestinal chronic bleeding`
We have revised the text (line 201).
Line 206: instead of `thus` write `it is therefore possible that the amount of RBCs destroyed by IMHA ... and that this did not cause overt signs of hemolysis such as hyperbilirubinemia, bilirubinuria or hemoglobinuria`
We have revised the text (lines 201-203).
Line 207: jaundice is the clinical syndrome that only is visible when bilirubin is markedly increased, so write hyperbilirubinemia instead of jaundice
We have revised the text (line 203).
Line 208: change `and` into `or`
We have revised the text (line 203).
Line 209: write `Pure red cell aplasia was however considered less likely because spherocytes ...
We have revised the text (line 203).
Line 210: remove `thus` and write `It was therefore considered more likely that the hemolysis had just begun`
We have revised the text (line 206).
Line 210: Write `The author`s think it is most likely that multiple mechanisms, including ...
We have revised the text (lines 206-207).
Line 214-221: this section is just not strong enough and confusing. I think the author`s either need to rewrite it and thoroughly discuss all the possible differential diagnosis for hypoalbuminemia, hypoproteinemia and hypoglycemia or write something in the line of `The hypoproteinemia and hypoalbuminemia initially was thought to be a consequence of gastro intestinal bleeding. However, signs such as melena or histopathologic evidence of transmural invasion of the tumor were not present. The presence of lymphangiectasia that was found on histopathology and was thought to be a consequence of physical obstruction of the lymphatic vessels by the tumor, could have caused some intestinal protein loss, however other causes for hypoproteinemia such as chronic inflammation are also considered possible.` And then proceed with `In addition, hypoglycemia ....`
We have revised the text (lines 220-226).
Line 222: write `due to anemia`
We have revised the text (line 229).
Line 223: remove `also`
We have removed.
Line 224-227: remove this section
We have removed this section.
Line 229: remove `was`
We have removed.
Reviewer 3 Report
Thank you for providing the revised version of the manuscript entitled “Complete remission of associative immune-mediated hemolytic anemia in a dog following surgical resection of intestinal leiomyosarcoma”. After reviewing the authors response and new version of the manuscript, I’m convinced this manuscript can be acceptable for publication in the current form. This reviewer has no further comments.
Author Response
Thanks for your comments